# Treatment with *Uncaria tomentosa* Promotes Apoptosis in B16-BL6 Mouse Melanoma Cells and Inhibits the Growth of B16-BL6 Tumours

**DOI:** 10.3390/molecules26041066

**Published:** 2021-02-18

**Authors:** Ali Zari, Hajer Alfarteesh, Carly Buckner, Robert Lafrenie

**Affiliations:** Department of Biology and Biomolecular Sciences, Laurentian University, Sudbury, ON P3E 2C6, Canada; ali_t_zarea@hotmail.com (A.Z.); hajeralfarteesh8787@gmail.com (H.A.); carlybuckner2@gmail.com (C.B.)

**Keywords:** *Uncaria tomentosa*, cancer, mouse melanoma, apoptosis, histochemistry, immune infiltration

## Abstract

*Uncaria tomentosa* is a medicinal plant native to Peru that has been traditionally used in the treatment of various inflammatory disorders. In this study, the effectiveness of *U. tomentosa* as an anti-cancer agent was assessed using the growth and survival of B16-BL6 mouse melanoma cells. B16-BL6 cell cultures treated with both ethanol and phosphate-buffered saline (PBS) extracts of *U. tomentosa* displayed up to 80% lower levels of growth and increased apoptosis compared to vehicle controls. Treatment with ethanolic extracts of *Uncaria tomentosa* were much more effective than treatment with aqueous extracts. *U. tomentosa* was also shown to inhibit B16-BL6 cell growth in C57/bl mice in vivo. Mice injected with both the ethanolic and aqueous extracts of *U. tomentosa* showed a 59 ± 13% decrease in B16-BL6 tumour weight and a 40 ± 9% decrease in tumour size. Histochemical analysis of the B16-BL6 tumours showed a strong reduction in the Ki-67 cell proliferation marker in *U. tomentosa*-treated mice and a small, but insignificant increase in terminal transferase dUTP nick labelling (TUNEL) staining. Furthermore, *U. tomentosa* extracts reduced angiogenic markers and reduced the infiltration of T cells into the tumours. Collectively, the results in this study concluded that *U. tomentosa* has potent anti-cancer activity that significantly inhibited cancer cells in vitro and in vivo.

## 1. Introduction

*Uncaria tomentosa (Willd.) DC (Rubiaceae*), more commonly known as Cat’s Claw, is a medicinal plant that has been traditionally used by the Aboriginal peoples of Amazonian Peru. *Uncaria tomentosa* is one of the most popular natural health products in North America and Europe [1] and is widely used by patients for its purported activities against inflammatory diseases such as arthritis, gastrointestinal disease, and viral infections as well as for the prevention or treatment of cancers [2,3,4].

Extracts of *Uncaria tomentosa* are able to potently inhibit acute inflammatory activity. We showed that the treatment of THP-1 monocyte-like cells with *Uncaria tomentosa* for 24 h can inhibit the ability of lipopolysaccharide (LPS) to increase TNF-α production (although IL-1β production is stimulated) [5,6] by inhibiting the activation of the NF-kB p52 subunit. Others have shown that the treatment of mice with *Uncaria* extracts (or components) can inhibit the production of pro-inflammatory cytokines in response to LPS treatment [7,8] and inhibit carrageenan-induced edema and inflammation by 40% [9]. Human clinical trials have also shown that treatment with an extract of *Uncaria tomentosa* can decrease some of the inflammatory symptoms of rheumatoid arthritis [10] or osteoarthritis [11].

Treatment with *Uncaria tomentosa* extracts has also been shown to have anti-cancer activity. In vitro treatment with *Uncaria tomentosa* extracts can decrease proliferation or induce apoptosis in a variety of cancer cells including leukemias [12,13,14], gliomas or neuroblastomas [15], colon cancer [16], bladder cancer [17,18], thyroid cancer [19], or breast cancer cells [20,21,22]. In vivo experiments have shown that *Uncaria tomentosa* can inhibit the growth of implanted tumours (B16-BL6 mouse melanoma or W256 rat choriocarcinoma) in rodents [8,23,24]. The intraperitoneal injection of an aqueous extract of *Uncaria* [25] or an ethanol extract resuspended in phosphate-buffered saline (PBS) [8] was shown to inhibit the growth of B16-BL6 lung tumours in a metastatic model (intravenous injection) by 70% and decrease the expression of TNF-α. The intraperitoneal injection of the resuspended ethanol extract was also able to inhibit the growth of subcutaneous “primary” B16-BL6 tumours by up to 75% [8]. However, the mechanisms for these changes are largely unknown. Clinical trials have shown that the oral consumption of water extracts of *Uncaria tomentosa* does not affect tumour growth in patients with breast or colorectal cancer being treated with standard chemotherapy [26,27,28,29]. However, patients in these trials have shown a significant decrease in the side effects resulting from chemotherapy such as neutropenia, malaise, inflammatory side effects, and DNA damage [26,27,28,29]. In some cases, the improvement experienced following chemotherapy was related to the anti-inflammatory activity of *Uncaria* [26], however, this was not consistent for all cases [30].

*Uncaria tomentosa* contains a large number of chemical components, including quinovic acid glycosides, triterpenes, and oxindole alkaloids [31,32], and differences in the methods used to produce extracts can result in different physiological effects. For example, an aqueous extract of *Uncaria tomentosa*, called C-Met-100 (which is rich in carboxyl alkyl esters but which contains relatively low levels of alkaloids [33]) was shown to be anti-inflammatory and immune system restorative [28,34,35], but did not provide significant anti-cancer activity in clinical trials [36]. Treatment with extracts enriched in quinovic acid glycosides were shown to decrease inflammatory side effects associated with anti-cancer drugs [29] and promote apoptosis in human bladder cancer cells [17]. A non-alkaloid phenolic fraction inhibited the growth of W256 tumours in rats [23]. However, alcohol extracts of *Uncaria tomentosa,* which contain a number of pentacyclic oxindole alkaloids, have been consistently shown to have cytotoxic effects against bladder [18], thyroid [19], cervical, and breast cancer cells [20,21].

This study shows that an ethanol extract of *Uncaria tomentosa*, which is rich in alkaloids, has potent pro-apoptotic effects on cultured B16-BL6 cells and on the growth of B16-BL6 tumours in C57bl mice. The aqueous extract of *Uncaria tomentosa*, which has relatively low levels of alkaloids, is less active in vitro but is still able to inhibit tumour growth in vivo.

## 2. Results

### 2.1. Uncaria tomentosa Extracts Inhibit Proliferation of B16-BL6 Cells

These experiments were designed to show the effect of *Uncaria tomentosa* on the growth of B16-BL6 cell cultures. Since previous results had shown differences in the effects between ethanolic and aqueous extracts, the effect of different extracts was tested. Treatment of B16-BL6 cells with the 70% ethanol extract of *Uncaria tomentosa* for 4 days showed an approximate 16 ± 12% decrease in cell viability at the lowest concentration of 4 µg/mL, an 82 ± 12% decrease in cell number at the medium dose of 40 µg/mL, and the complete inhibition of growth at the highest dose of 100 µg/mL (Figure 1A). Treatment of B16-BL6 cells with *Uncaria tomentosa* extracted with phosphate-buffered saline (PBS) was less effective and treatment with the low dose did not significantly inhibit B16-BL6 cell viability although treatment with the high dose inhibited B16-BL6 cells by greater than 84 ± 14%. Non-malignant NIH3T3 and C2C12 mouse cells treated with *Uncaria tomentosa* extracts showed a lower level of sensitivity. Treatment with the highest concentration of the ethanolic extract for 4 days inhibited the growth of NIH3T3 cells by 32 ± 12% and C2C12 cells by 27 ± 9% while treatment with the highest dose of the PBS extract inhibited the growth of NIH3T3 cells by 18 ± 8% and C2C12 cells by 11 ± 7% (Figure 1B). These results showed that there were differences in the sensitivity of the different cell lines to *Uncaria tomentosa* extracts in a dose-dependent manner.

B16-BL6 cells treated with low (4 µg/mL), medium (40 µg/mL), and high (100 µg/mL) doses of the ethanol extract of *Uncaria tomentosa* extracts showed dose-dependent changes in cell shape and confluence using phase contrast microscopy (Figure 1C). Cells treated with the low dose showed a small decrease in cell density after 3 days of treatment although the shape and size of the cells were similar to media-treated controls. A significant decrease in the confluency of cell cultures was observed following the treatment with the medium (56 ± 11%) and the high doses (72 ± 12%) of the ethanol extract of *Uncaria tomentosa* for 3 days.

The proportion of cells expressing the Ki-67 proliferation nuclear antigen in cultured B16-BL6 cells was also shown to be decreased in cells treated with *Uncaria tomentosa* extracted with ethanol or PBS compared to media-treated controls in a dose-, time- and extract-dependent manner. Specifically, treatment of B16-BL6 cells with the highest dose of *Uncaria tomentosa* extracted with ethanol decreased the number of Ki-67-stained cells at 48 h post-treatment by 98 ± 2%, similar to treatment with the positive control, Camptothecin (Figure 1C). In contrast, B16-BL6 cells treated with the highest dose of the PBS extract of *Uncaria tomentosa* for only 24 h did not significantly inhibit Ki-67 staining although treatment for 48 h did decrease the number of Ki-67-stained cells by 41 ± 13%. These results are consistent with the results from the cell viability experiments.

### 2.2. HPLC Characterization of Uncaria tomentosa Extracts

Since there were significant differences between the ethanolic and PBS extracts in terms of cell viability, the analysis of the extracts was examined. *Uncaria tomentosa* bark powder was extracted by boiling in 70% ethanol or PBS, pH 7.4 for use in experiments and analysis by HPLC. Extracts were analyzed on a C18 column and peaks detected at a wavelength of 245 nm and compared to molecular standards corresponding to known components of *Uncaria tomentosa*. The ethanol extracts were shown to include the alkaloids, uncarine D, mitraphylline, uncarine C, isomitraphyline, rhynophyline, and uncarine F, while a significantly lower amount of these alkaloids was present in the PBS extract (Figure 2A). The analysis of the area under each of the absorbance peaks shows that, on average, the ethanol extracts contained the compounds at 0.2–2.6 mg/mg of plant material and that there was about 10-fold less of each of these alkaloids in the PBS extract.

The *Uncaria tomentosa* extracts were fractionated into different ethanolic fractions, as described in the Materials and Methods Section, in order to enrich for the potential active molecules and then test for their effect on the proliferation of B16-BL6 cells. HPLC showed that the level of the various oxindole alkaloids varied in each of the fractions with the highest levels, in particular uncarine D, mitraphylline, and uncarine C, in the 40 and 60% ethanol fractions (Figure 2B). The treatment of the B16-BL6 cell line with 1 µg/mL of each reconstituted fraction in culture media showed a significant decrease in cell viability for the 40 and 60% fractions using an 3-(4,5-dimethylthiazol-2-yl)-2,5-diphenyltetrazolium bromide (MTT) assay over 1 to 4 days in the continued presence of the extract (Figure 2C).

### 2.3. Treatment with Uncaria tomentosa Extracts Induces Apoptosis in B16-BL6 Cells

Since B16-BL6 cells treated with *Uncaria tomentosa* extracts showed a significant decline in viability, the effects of *Uncaria* extracts on cell death and apoptosis were examined. B16-BL6 cells treated with the high (100 µg/mL) dose of the ethanol extract of *Uncaria tomentosa* for 3 days showed changes in cell and nuclear morphology consistent with apoptosis using the fluorescence microscopy of cells stained with acridine orange and ethidium bromide (Figure 3A). Cells treated with the PBS extract or with the low dose of the ethanol extract showed a similar shape and size compared to media-treated controls. This corresponded to ethidium bromide staining of <20% of cells treated with the PBS extracts but 77 ± 11% of the cells treated with the highest dose of *Uncaria tomentosa* stained with ethidium bromide corresponding to an increase in late apoptotic morphology which was similar to Camptothecin-treated cells.

A terminal transferease dUTP nick labelling (TUNEL) assay was performed to detect DNA fragmentation, a hallmark of apoptosis. Treatment with the highest dose of the ethanol extract of *Uncaria tomentosa* for 2 days induced DNA fragmentation in approximately 83 ± 12% of the cells, similar to Camptothecin-treated cells (Figure 3B). Cells treated with the PBS extracts of *Uncaria tomentosa* or the low level of the ethanol extract induced TUNEL staining in <15% of the treated cells after 48 h. Treatment with *Uncaria tomentosa* extracted with ethanol was more effective at inducing apoptosis when compared to *Uncaria* extracted in PBS. Results from the TUNEL staining correlated with the results obtained from the cell viability experiments.

Flow cytometry analysis showed that the treatment of B16-BL6 cells with the ethanol extract of *Uncaria tomentosa* extracts increased the presence of sub-G1 apoptotic cells in cell cycle analysis. B16-BL6 cells treated with the high dose of the ethanol extract of *Uncaria tomentosa* for 48 h induced 78.4% of the cells in the sub-G1 peak indicating a large increase in apoptosis (Figure 3C). Treatment for 24 h increased the sub-G1 population to 61.9%. This is similar to treatment with Camptothecin which showed an increase up to 74.9% of the cells in the sub-G1 peak. The treatment of the cells with the low dose of the ethanol extract of *Uncaria tomentosa* or the PBS extract for 24 or 48 h did not significantly increase the number of cells in the sub-G1 peak or alter the number of cells in the different phases of the cell cycle.

The effect of *Uncaria tomentosa* extracts on cell signaling pathways that are related to cell growth and apoptosis were also examined for a role in potential Uncaria-mediated effects. The treatment of B16-BL6 cells with low (4 µg/mL), medium (40 µg/mL), and high (100 µg/mL) doses of *Uncaria tomentosa* inhibited the ERK and Akt signaling pathways in a dose- and extract-dependent manner as determined by the immunoblot analysis of ERK and Akt phosphorylation. Treatment with the highest dose of the ethanol extract decreased ERK (threonine 202/204) phosphorylation to 20% of untreated controls and decreased Akt (serine 473) phosphorylation to 12% of untreated controls (Figure 4A). This was supported by the pathway analysis of transcription factors using the Cignal transcription factor array system that showed a 50% decrease in the MAP kinase signaling pathway and a 60% decrease in the Akt/PI3 kinase pathway in B16-BL6 cells treated with the high dose of the ethanol extract of *Uncaria tomentosa* (Figure 4B). Treatment also decreased the transcriptional activity of the MEF2, c-myc and PPARγ pathways by 30–40% which are all linked to the activation of the MAP kinase and/or Akt pathways and have been linked to oncogenic behaviour. Treatment with *Uncaria tomentosa* increased the transcriptional activities associated with estrogen (4.4-fold) and androgen (1.7-fold) receptor signaling as well as with the STAT3 and TGF-β pathways (1.7-fold) while several pathways including the cell cycle, p53, hedgehog, Nanog, retinoic acid, and Wnt pathways were not significantly affected by Uncaria tomentosa treatment.

Immunoblot analysis also showed that treatment of B16-BL6 cells with *Uncaria tomentosa* extracts increased the cleavage of caspase-3, another marker of apoptosis (Figure 4A). Treatment with the PBS extracts of *Uncaria tomentosa* for 48 h increased the level of caspase-3 cleavage by up to 1.5-fold while treatment with the highest dose of the ethanol extract increased caspase-3 cleavage by 2.6-fold, which was similar to treatment with Camptothecin.

### 2.4. Uncaria tomentosa Extracts Inhibit B16-BL6 Tumour Growth in C57BL/6 Mice

The effect of *Uncaria tomentosa* extracts was tested on the growth of B16-BL6 tumours in C57BL/6 mice in two separate experiments. For these experiments, *Uncaria tomentosa* was extracted in 70% ethanol or PBS, freeze-dried, and then resuspended in PBS for injection. Biweekly injection of *Uncaria tomentosa* extracts significantly inhibited the growth of B16-BL6 tumours compared to vehicle only controls. Mice treated with *Uncaria tomentosa* extracted with both ethanol and PBS showed a 59 ± 13% decrease in tumour weight (*p* < 0.001) for the combined two experiments (Figure 5A). There were no significant differences in these effects between the two experiments (Figure 5B). Tukey analysis showed significant differences between the vehicle control and each of the four treated groups (*p* < 0.05) but there were no significant differences between the four treated groups. Tumour size (diameter) was decreased by 40 ± 9% (*p* < 0.05) in mice treated with both the ethanol and PBS extracts of *Uncaria tomentosa* compared to the control animals, as determined using two-way ANOVA and Tukey tests. In addition, there were no significant major effects or interactions for the two levels of treatment (injection type: intraperitoneal versus intratumour; and extract type: ethanol versus PBS) found using two-way ANOVA. There were no significant differences in mouse body weight between the five groups supporting the idea that the injection of *Uncaria tomentosa* did not significantly impact mouse health.

### 2.5. Histochemical Analysis Was Performed for B16-BL6 Tumours

The B16-BL6 tumours isolated from the mice were subjected to histochemical staining to measure the effect of treatment with *Uncaria tomentosa* on cell growth, cell death and immune cell markers. The level of Ki-67 staining was determined using an ordinal scale (0–5 based on staining intensity) in each of the tumour sections and scored for the average for each mouse. The staining intensity for Ki-67 in tumours isolated from mice that had been treated with vehicle, showed a significantly higher staining score (25.2 ± 1.4) when compared to tumours from mice treated with either the ethanol (17.1 ± 1.2) or the PBS extract of *Uncaria tomentosa* (17.1 ± 1.6) which were not different from each other (Figure 6). The Ki-67 antigen has been used as a marker of actively proliferating cells and examination of the sections at high magnification confirmed that the positive staining was associated with the nucleus of the cells as expected.

The levels of the TUNEL staining reaction were examined as a measure of cellular apoptosis within the tumour tissue. The average staining intensity for the number of TUNEL positive cells showed that staining was lowest in tumours from animals injected with only the vehicle (22.7 ± 3.5); however, this was not significantly different from the level of staining in the tumours from mice treated with *Uncaria tomentosa* extracted with ethanol (27.9 ± 3.5) or PBS (26.6 ± 3.1). The examination of the sections at high magnification indicated that positive staining was consistent with localization in the nucleus.

Staining for Factor VIII, a marker of endothelial cells in the tumour sections, was conducted as a measure of angiogenesis. The average staining intensity for Factor VIII was greater in the tumours from mice treated with the vehicle (24.3 ± 1.3) compared to tumours from mice treated with the ethanol (17.4 ± 1.6) or PBS (18.2 ± 2.0) extracts of *Uncaria tomentosa*. Factor VIII staining was frequently associated with the presence of blood vessels in the tumour cross-section but in some sections, staining was more diffuse which may indicate weakly organized vascular beds within the growing tumour.

The number of endothelial cells in the tumour sections per high power microscope field (approximately 0.04 mm^2^), was also determined by staining with an anti-PECAM-1 antibody (anti-PECAM-1-Alexa-488). The number of cells/high power field (hpf) was lower (*p* < 0.05) in mice treated with *Uncaria tomentosa* extracts compared to vehicle only-treated mice, consistent with the results of the Factor VIII-stained samples. However, there did not appear to be a difference in the number of cells that expressed inflammatory markers on the endothelial cells. The number of ICAM-1 and E-selectin-stained cells (anti-ICAM-1-Alexa-633 and anti-E-selectin-Alexa-546), which are markers of inflammation, was lower than the number of PECAM-1-stained cells (in the same tissue sections) and not different between tumour sections from mice treated with *Uncaria tomentosa* extracts or vehicle only.

The histochemical sections were also stained with antibodies that recognize various immune cell markers and the number of stained cells/high-power field (hpf) in each section was counted to measure the relative levels of immune cell infiltration into B16-BL6 tumours treated with extracts of *Uncaria tomentosa*. The infiltration of T and B cells into tumours has been shown to correlate with the outcome [8]. The number of tumour-infiltrating cells stained with the T cell marker CD3 (anti-CD3-Alexa-546), the T helper cell marker, CD4 (anti-CD4-Alexa-488), and the cytotoxic T cell marker CD8 (anti-CD8-Alexa-633) all showed a small but significant decrease in immune infiltration into tumours isolated from mice treated with *Uncaria tomentosa* extracts compared to animals injected with the vehicle only (Figure 7). The number of CD3 T cells in *Uncaria tomentosa*-treated mice was decreased to 374 ± 31 or 369 ± 40 cells/hpf for the PBS and ethanol extracts, respectively, from 434 ± 29 cells/hpf in the vehicle only-treated mice (*p* < 0.05). Similarly, the number of CD4 cells was decreased (*p* < 0.05) from 334 ± 22 cells/hpf to 276 ± 30 or 272 ± 35 cells/hpf in mice treated with the PBS or ethanol extracts, respectively. The number of leukocytes infiltrating the tumours, determined by staining with the pan leukocyte marker CD45 (anti-CD45-Alexa 633), was not different in tumours from mice treated with extracts of *Uncaria tomentosa* or vehicle. Similarly, the number of B cells infiltrating the tumours, as determined by staining with CD19 (anti-CD19 Alexa-549) was not different in tumours from mice treated with extracts of *Uncaria tomentosa* or vehicle.

## 3. Discussion

Treatment of B16-BL6 mouse melanoma cells in culture with a 70% ethanol extract of *Uncaria tomentosa* was able to significantly inhibit cell viability and induce apoptosis. Furthermore, intraperitoneal injections of both the aqueous and 70% ethanol extracts of *Uncaria tomentosa* (twice weekly) inhibited the growth of ectopic B16-BL6 tumours in C57BL mice by approximately 60%.

The decrease in B16-BL6 cell viability and the increase in cellular apoptosis was dependent on the type of extract (ethanol versus PBS), the dose, and the duration of treatment. For example, the treatment of B16-BL6 cells with 100 µg/mL of the ethanol extract of *Uncaria tomentosa* for more than 3 days almost completely blocked cell viability and increased apoptosis to 80% of treated cells. Lower doses or shorter durations were both less effective. Furthermore, treatment with 100 µg/mL of the PBS extract for 3 days only decreased viability and increased apoptosis by <20% of the treated cells. This suggests that the ethanol extract contains a higher level of a compound or compounds that promote cell death.

*Uncaria tomentosa* treatment appears to distinguish between B16-BL6 cancer cells and non-cancer cells. The viability of B16-BL6 cells appears to be inhibited to a greater degree than NIH3T3 or C2C12 cells, although this might be due to the fact that faster growing cells might be more affected than slower growing cells in vitro. However, our in vivo mouse experiments also did not show significant effects on mouse health supporting the idea that *Uncaria tomentosa* is more effective at inducing apoptosis in malignant cells although a more systematic analysis of different cell types is required.

There is relatively little literature related to the activation or inhibition of signaling pathways following treatment of cells with *Uncaria tomentosa* extracts. Treatment of B16-BL6 cells with the 70% ethanol extract of *Uncaria tomentosa* was able to inhibit the phosphorylation of the ERK and Akt signaling molecules to a greater extent than the PBS extract. The decreased activity of the MAP kinase (ERK/MEK) and Akt/phosphoinositol-3 kinase and other related downstream pathways (PPARγ, MEF2, and c-myc [37]) was also shown by the transcription factor profiling assay. ERK phosphorylation and activity are linked to cell proliferation in response to growth factors or oncogenic activation and therefore the inhibition of ERK signaling is often linked to the inhibition of cell proliferation [38]. Akt phosphorylation and activity is linked to pathways that inhibit cellular apoptosis and the inhibition of Akt phosphorylation has been linked to an increase in cellular apoptosis [39]. Therefore, the ability of treatment with extracts of *Uncaria tomentosa* to inhibit ERK and Akt may be mechanistically related to the changes in cell viability and apoptosis; however, further experiments are required to test this possibility.

One of the current issues with *Uncaria tomentosa* research is that the identity of the active components in the extracts is still unclear. Different extraction methods have produced extracts with differing efficacy and multiple individual components (or enriched fractions) have been shown to mediate the same activity [40]. We found that treatment with ethanol extracts of *Uncaria tomentosa*, which contained significant amounts of pentacyclic oxindole alkaloids, was highly active at inhibiting cell viability and inducing cellular apoptosis and that aqueous extracts were less active. HPLC analysis showed that the ethanol extract contained almost 10-fold higher levels of the diagnostic *Uncaria* alkaloids, uncarine D, mitraphylline, uncarine C, isomitraphyline, rhynophyline, and uncarine E, when compared to the PBS extracts. Others have shown that ethanol extracts are rich in pentacyclic oxindole alkaloids [5,6,7,8,9,10] and are important in mediating both anti-inflammatory and anti-cancer activities [12,17,29,33]. Some individual alkaloids, isolated from *Uncaria* extracts, such as mitroaphylline, have anti-proliferative [15] or pro-apoptotic effects on cancer cells [22] while Uncarine C (pteropodine) can inhibit DNA damage and scavenge free radicals in doxorubicin-treated mice [41]. Thus, the alkaloids are the best candidates for anti-cancer activity. However, an aqueous extract of *Uncaria tomentosa*, called C-Met-100 (which is rich in carboxyl alkyl esters but contains low levels of alkaloids) is anti-inflammatory and an immune system restorative [35,42]. C-Med 100, is rich in quinovic acid and quinovic acid glycoconjugates, suggesting thar these molecules can inhibit inflammatory responses in vitro [34], promote lymphocyte survival in mice [35,42], and enhance DNA repair activity [36,43] independent of alkaloids [33]. The C-Med100 extract was able to inhibit cancer cell growth in vitro [44] and in mouse models of B16-BL6 metastasis [25] suggesting that these molecules might also have anti-cancer activity.

Fractionation of the ethanol extracts showed that alkaloid-rich fractions have stronger anti-viability activity on B16-BL6 cells. Fractions with higher levels of alkaloids following polyvinylpyrrolidone-column fractionation (which also removes polyphenols [45]) were shown to have higher levels of activity in inhibiting B16-BL6 viability. Furthermore, the ethanol extracts that contained higher levels of alkaloids were more active at inhibiting B16-BL6 viability than aqueous extracts which contain low levels of alkaloids. These results show a correlation between the level of oxindole alkaloids and anti-cancer activity although further experiments are required to determine if a specific molecule, or a combination of molecules, is required to mediate anti-cancer activity.

We and others [8] have shown that the intraperitoneal injection of *Uncaria tomentosa* extracts can inhibit the growth of syngeneic B16-BL6 melanoma tumours in immune-competent C57BL mice. Interestingly, Fazio et al. [8] used an *Uncaria tomentosa* extract which did not significantly inhibit cancer cell growth in vitro (B16-BL6, Hela, or A549 cells) but was able to decrease B16-BL6 tumour size and inhibit inflammatory cytokine expression. In our experiments, mice were injected with *Uncaria tomentosa* extracts either intraperitoneally or directly into the growing tumour with 20 mg/mouse of a reconstituted *Uncaria tomentosa* extract twice weekly. Under these conditions, treatment with both the ethanol and PBS extracts inhibited tumour growth by approximately 60% compared to mice injected only with vehicle. The extracts used in our study were able to induce apoptosis in cultured cancer cells (including B16-BL6 cells), although there were significant differences between the ethanol and PBS extracts, which suggest that repeated dosage with a lower concentration than is required to inhibit tumour growth in vitro is effective in vivo.

Consistent with our results, animal experiments have shown that *Uncaria tomentosa* is relatively non-toxic: the LD50 in rats is >8 g/kg [43] and in mice is >16 g/kg [3]. This indicates that its activity is not likely simply mediated by generalized toxicity. The ability of extracts of *Uncaria tomentosa* to inhibit the growth of tumours in mice without having any notable negative effects on normal mouse tissues or mouse “health” seems somewhat at odds with the observation that *Uncaria tomentosa* extracts can also inhibit viability and promote apoptosis in non-malignant cells. There are a few possible explanations which include the idea that extracts primarily affect rapidly proliferating cells both in vitro and in vivo, that there is some unique feature of tumours (e.g., vasculature or hypoxic environment) which might enhance “drug” deposition, or that cancer cells in a tumour are somehow more sensitive than cancer cells in monolayer cultures. It is also possible that the anti-tumour activity is a result of combined cytotoxic effects and immune-mediated responses to *Uncaria tomentosia* extracts. Consistent with the in vitro results, histochemical analysis of the B16-BL6 tumours showed a significant decrease in staining with the Ki-67 proliferation antigen in mice treated with *Uncaria tomentosa* extracts. This suggests that treatment with Uncaria extracts has an anti-proliferative effect in vivo. However, the TUNEL analysis of B16-BL6 tumours was not significantly different between mice treated with *Uncaria* extracts or vehicle only-treated animals, which is different from the in vitro results. Since this analysis was performed on the animals at the endpoint of the experiment, it is possible that any *Uncaria*-specific effects could have been obscured by increased apoptosis in the larger, hypoxic tumours in the vehicle only-treated animals.

Tumours treated with extracts of *Uncaria tomentosa* showed a lower level of angiogenesis compared to vehicle-treated tumours as indicated by the decreased staining of the tumour sections for Factor VIII or PECAM-1 (CD31) [46], which are both markers of endothelial cells. The lower level of angiogenesis in *Uncaria*-treated animals is associated with the smaller tumours and could be involved in the differences in tumour growth. A lower number of cells stained with the endothelial inflammation markers, ICAM-1 or E-selectin [47], compared to PECAM-1 and there were no differences between *Uncaria*- and vehicle only-treated mice the time of sacrifice. This indicates that cellular inflammation within the tumour microenvironment does not contribute to the late-stage effects of *Uncaria* on tumour growth.

The number of tumour-infiltrating cells stained with the T cell markers CD3 (anti-CD3-Alexa-546), CD4 (anti-CD4-Alexa-488), and CD8 (anti-CD8-Alexa-633) were all decreased by about 10% in tumours isolated from mice treated with *Uncaria tomentosa* extracts compared to animals injected with vehicle only. While this is a small difference in cell number at sacrifice, the differences in T cell number do indicate a potential effect of *Uncaria* treatment on immune infiltration into the tumours which could affect tumour growth. The number of leukocytes infiltrating the tumours, determined by staining with the pan leukocyte marker CD45 (anti-CD45-Alexa 633), or the number of B cells detected with CD19 antibodies (anti-CD19 Alexa-549), was not different in tumours from mice treated with extracts of *Uncaria tomentosa* or vehicle suggesting a specific infiltration of immune cells into the tumour.

The results of these studies show that the treatment of B16-BL6 cancer cells or the injection of mice bearing B16-BL6 cancers with *Uncaria tomentosa* extracts can inhibit tumour cell growth. The in vivo studies indicate that *Uncaria tomentosa* is well tolerated and might have potential as a cancer therapy. For example, there might be a place for *Uncaria* in the treatment of patients with malignant melanoma since current therapies largely involve specific immunotherapy or chemical inhibitors of signaling molecules (like B-Raf) which are effective only in a subgroup of patients. Conventional chemotherapy, such as dacarbazine and temozolomide, is not very effective in the long-term treatment of malignant melanoma and there might be a benefit to adding components of the *Uncaria* extract to this treatment. The few human clinical trials which have been conducted (commonly using 250–350 mg of ethanol extract/day), show that *Uncaria* is well tolerated, although there were some side effects reported in patients with renal disease [48] or Parkinson’s disease [49]. These trials are mostly related to minimizing the side effects of chemotherapy treatment in cancer patients or in patients with chronic inflammatory disorders [26,27,28,29] with some studies reporting improvements in patient quality of life or decreased neutropenia. None of the trials involving breast or colorectal cancer patients have shown effects on cancer progression related to *Uncaria tomentosa* treatment—however, these trials were not designed to test that endpoint. In addition, the *Uncaria tomentosa* used in the clinical trials was provided orally as a dried preparation which could affect the available concentration and bioavailability of many components. These studies suggest future work to identify a particular compound (or extract) from *Uncaria tomentosa* with a logical mechanistic link to the induction of cellular apoptosis which would be valuable in designing a new therapeutic.

## 4. Materials and Methods

### 4.1. Cell Lines and Tissue Culture

B16-BL6 (murine melanoma) [50], NIH3T3 fibroblasts (which are immortal but not malignant), and C2C12 cells (which are immortal, non-malignant mouse myoblasts) (obtained from the American Type Culture Collection, Manassas, VA, USA), were maintained in Dulbecco’s modified essential medium (DMEM, Hyclone, Logan, UT, USA) supplemented with 10% fetal bovine serum (Hyclone), 100 µg/mL streptomycin, and 100 U/mL penicillin (Invitrogen, Burlington, ON, Canada). The cells were cultured at 37 °C in 5% CO_2_.

### 4.2. Uncaria tomentosa Extracts and Characterization

*Uncaria tomentosa* was obtained from Rosario Rojas (Lima, Peru) as a dried powder prepared from the bark or was purchased as a natural product supplement (Cat’s Claw extract, Now Foods, Bloomington, IL. code 84618, USA). The dried plant product was extracted with either PBS, pH 7.4, or 70% ethanol and the soluble fraction used in the current research. Extracts were prepared by suspending 20 g of *Uncaria tomentosa* powder in 200 mL of PBS, pH 7.4, or in 200 mL of 70% ethanol and heated to a slow boil with refluxing for 1 h. The mixture was cooled, the bark debris removed by centrifugation, and the supernatant filtered using a 0.22 μm syringe filter. The extract was stored in aliquots at −80 °C. For both the animal experiments and the chemical fractionation experiments, the *Uncaria tomentosa* extracts were freeze-dried, the powder weighed, and resuspended in water immediately prior to use for the experiments.

The 70% ethanol extract of *Uncaria tomentosa* was fractionated over an ethanol gradient. For this experiment, 100 mL of 70% ethanol extract was freeze-dried and resuspended in 100 mL water. The resuspended extract was applied to a polyvinylpolypyrollidone (PVPP) column [45] and fractionated into 100 mL fractions as follows: a water fraction, 20% ethanol fraction, 40% ethanol fraction, 60% ethanol fraction, 80% ethanol fraction, and 95% ethanol fraction. Each fraction was freeze-dried and resuspended in 70% ethanol at 10 mg/mL of dried powder. The PVPP fractions were analyzed by HPLC and examined for effects against the B16-BL6 cancer cells using MTT assays and compared to both the ethanol and PBS whole *Uncaria tomentosa* extracts.

### 4.3. High-Performance Liquid Chromatography (HPLC)

The *Uncaria tomentosa* extracts or PVPP fractions were filtered through a 0.22 um syringe filter and then analyzed by HPLC on a Breeze 2 chromatography system (Waters Inc, Toronto, ON, Canada) fitted with a Sunfire C18 column 3.5 μm resin 4.6 × 100 mm. A gradient mobile phase was pumped onto the column at a flow rate of 1 mL/min. A linear gradient was formed by mixing the solvents starting with 100% buffer A (60 volumes 10 mM phosphate buffer, pH 6.6, 20 volumes acetonitrile, and 20 volumes methanol) and finished with 100% B (30 volumes 10 mM phosphate buffer, pH 6.6, 35 volumes acetonitrile, and 25 volumes methanol) over a period of 40 min [51]. Then, 100% buffer B were pumped onto the column for 10 min and lastly a gradient starting with 100% B and finishing with 100% A were pumped through the column for 5 min. The column was injected with a 5 µL aliquot of each sample at the beginning of each run. Components in the *Uncaria tomentosa* extracts were detected at 245 nm and peak areas calculated to determine the relative amount of each component. Various peaks were identified by comparison to a series of *Uncaria tomentosa* standards (1 mg/mL of each standard) using the same conditions for HPLC characterization. These standards included uncarine D, mitraphylline, uncarine C, isomitraphyline, rhynophyline, and uncarine E (ChromaDex, Irvine, CA, USA).

### 4.4. Methyl Tetrazolium Blue Viability Assay

The viability of the cells was measured using the methyl tetrazolium (MTT) blue reduction assay. The cells were plated on 96-well plates at 2 × 10^3^ cells/well on day 0 and then treated with low (4 µg/mL) to high (100 µg/mL) *Uncaria tomentosa* extracts on day 1. Each day, a replicate plate was sampled: 5 µL/well of a 5 mg/mL MTT solution was added and incubated for 3 h; the medium was removed and the cells solubilized in 100 µL DMSO; and the absorbance was measured at 540 nm. Each experiment was performed using 6 wells/condition and the average was determined. The percent inhibition of growth on each day was determined as the % decrease in absorbance compared to the cells treated only with media and the mean percentage inhibition and standard deviation reported for 3 independent experiments.

### 4.5. Flow Cytometry

Flow cytometry was performed to measure the DNA content in treated cell lines using the propidium iodide dye. The relative proportion of cells in the Sub-G1 (apoptosis peak), G1, S (DNA replication), and G2/M (cell checkpoint and mitotic phases) under the influence of *Uncaria tomentosa* extracts was determined. B16-BL6 cell cultures were treated with *Uncaria tomentosa* extracts at 4, 40, or 100 μg/mL (or with 6 μg/mL of Camptothecin as a positive control for apoptosis) for 24 and 48 h. The cells were harvested and fixed in cold 70% ethanol. For staining, the cells were washed in PBS, pH 7.4, and suspended in propidium iodide (PI) staining solution (150 mM NaCl, 0.1% Triton 100, 5.0 mg/mL RNase A, and 20 μg/mL of propidium iodide) and incubated for at least 30 min, as described previously [52]. Cell cycle analysis was performed using a Beckman Coulter LS600 flow cytometer and cell profiles created. The % cells shown for each condition were calculated as the mean percent positive cells from 3 independent experiments.

### 4.6. Fluorescence Microscopy—Vital Staining

Cell morphology was examined in B16-BL6 cells, plated overnight on glass coverslips, treated with media or media containing low (4 μg/mL) and high (100 μg/mL) *Uncaria tomentosa* extracts or 10^−6^ M Camptothecin for 24–48 h. Cells were incubated in 10 μg/mL acridine orange and 10 μg/mL ethidium bromide for 15–30 min and visualized on a LSM5 Zeiss fluorescence microscope for apoptotic morphology. The percentage of cells stained with ethidium bromide compared to acridine orange-stained cells was determined from 3 microscope fields/experiment from 3 different experiments.

### 4.7. Fluorescence Microscopy—TUNEL

The TUNEL assay was used to detect DNA fragmentation using the Roche Diagnostics kit, as described [53]. The cells were plated on glass coverslips and then treated with media or media containing low (4 μg/mL) and high (100 μg/mL) *Uncaria tomentosa* extracts or 10^−6^ M Camptothecin (positive control for apoptosis) for 24–48 h. The coverslips were fixed in 10% formaldehyde solution, permeabilized in 1% Triton X-100 in PBS, and then incubated with 50 μL of the TUNEL reagent (Roche Diagnostics, Laval, QB, Canada) for 60 min at 37 °C. The coverslips were mounted on glass slides and visualized using an LSM5 Zeiss fluorescence microscope. The percent of positively stained cells is shown for at least 5 independent microscope fields (>100 total cells counted/condition).

### 4.8. Immunoblot Analysis

B16-BL6 cells were treated with media, low (4 μg/mL), medium (40 μg/mL), and high (100 μg/mL) doses of *Uncaria tomentosa* extracts, or 10^−6^ M Camptothecin for 48 h. The cells were harvested and lysed in RIPA buffer (1% Triton X-100, 0.5% SDS, 0.5% sodium deoxycholate, 150 mM sodium fluoride, 1 mM sodium orthovanadate) containing protease inhibitors (Roche Diagnostics). Total cell lysates were subjected to electrophoresis on 10% polycrylamide gels containing SDS and electrophoretically transferred to nitrocellulose membranes (Schleicher and Schuell, Xymotech Biosystems, Toronto, ON, Canada). The membranes were blocked by incubation in 5% bovine serum albumin (BSA) in Tris-buffered saline, pH 7.5 and 0.1% Tween-20 (TBST) and then incubated with antibodies against ERK phosphate (#9191, Cell Signaling Technology, Danvers, MA, USA), ERK1 and ERK2 (Santa Cruz Biotech., Santa Cruz, CA, USA), Akt phosphate (#9271, Cell Signaling Technology), Akt (#9272, Cell Signaling Technology), β-tubulin (Santa Cruz Biotech.), and caspase-3 (#9668, Cell Signaling Technology), in 0.5% BSA in TBST. The filters were washed and incubated with appropriate anti-IgG-horseradish peroxidase (HRP) conjugates (Santa Cruz Biotech., Santa Cruz, CA, USA) and the HRP detected by incubation in Supersignal Reagent (Pierce Chemical Co., Rockford, IL, USA) and exposed to Hyperfilm-ECL X-ray film (Amersham-Pharmacia, Oakville, ON, Canada). Densitometry was performed using AlfaEaseFC software (Protein Simple, San Jose, CA, USA) and fold changes in band intensity were compared to untreated cells and reported for 3 independent experiments.

### 4.9. Profiling Signaling Pathways Using Promoter–Reporter Assays

The Cignal 45-Pathway–Reporter Array (Qiagen, Toronto, ON, Canada) [54] was used to measure the relative activity of multiple signaling pathways in B16-BL6 cells treated with 100 µg/mL of the ethanol extract of *Uncaria tomentosa*. In brief, the cells were transfected with two plasmids, a firefly luciferase plasmid linked to pathway-specific promoter elements and a constitutively expressing *Renilla* luciferase plasmid, using Attractine transfection reagent. The cells were treated with the *Uncaria* extract or control media for 24 h, and the amount of the two luciferase reporters was measured using the Dual-Glo Luciferase Assay System (Promega, Madison, WI, USA). The relative level of luciferase was determined by subtracting the luminescence of the negative control plasmid and dividing by the *Renilla* luminescence level in each sample. The effect of *Uncaria tomentosa* treatment was determined by dividing the luminescence for the *Uncaria*-treated samples by the media-treated control samples for 2 independent experiments.

### 4.10. Animal Experiments

The B16-BL6 melanoma/C57bl/6 isogenic tumour transplantation model was performed to determine the anti-cancer activities of *Uncaria tomentosa* extracts in mice. A total of 50 syngeneic male C57BL/6 mice (8 weeks old) were used for two experiments (Charles River, Pointe-Claire, QB, Canada). In particular, 25 male mice were used for each experiment (experiment 1 and experiment 2) during a period of 18–20 days as described previously [55]. *Uncaria tomentosa* was extracted into 70% ethanol or PBS, pH 7.4, lyophilized to produce a powder, and then resuspended in PBS, pH 7.4 for injection to prevent injecting ethanol into the mice. All animal experiments were approved by the Laurentian University Animal Care Committee, protocol 2013-09-01, in accordance with the guidelines established by the Canadian Council for Animal Care.

Tumours were induced by the injection of 2 × 10^5^ cells B16-BL6 cancer cells, suspended in 100 µL of PBS, subcutaneously in the right flank of all of the mice. Animals were randomly separated into five equal groups (5 mice/group) in each experiment and the experiment was repeated: group 1, control animals injected intraperitoneally with PBS, pH 7.4; group 2, animals injected intraperitoneally with 100 μL of a 200 μg/mL PBS extract of *Uncaria tomentosa*; group 3, animals injected directly into the tumour with 100 μL of a 200 μg/mL PBS extract; group 4, animals injected intraperitoneally with 100 μL of a 200 μg/mL ethanol extract of *Uncaria tomentosa*; and, group 5, animals injected directly into the tumour with 100 μL of a 200 μg/mL ethanol extract. Injections of the *Uncaria* extracts were performed twice weekly starting 3 days after infection of the B16-BL6 cells: all animals received 4 injections of the extracts except for the animals in the intra-tumour group in experiment 1 which received only 2 injections of the extract in week two when the tumours were palpable. The animals were sacrificed on days 18–20 and all tumours were excised and collected. The weights of each mouse and the weight and average diameter of each tumour was recorded. For statistical analysis, two-way analysis of variance (ANOVA) and post-hoc Tukey tests (SPSS for windows, version 18, IBM, Markham, ON, Canada) were considered significant at *p* values of <0.05 for each experiment or for the combination of both experiments. The tumours were then fixed by incubation in EFA (720 mL ethanol (100%), 180 mL distilled H2O, 50 mL glacial acetic acid, and 50 mL formaldehyde).

### 4.11. Histochemical Analysis of Tissue Sections

Histochemical analysis of the isolated tumours was performed using several immune, cell growth, and cell death markers to examine the impact of treatment on B16-BL6 tumours. Samples were processed, embedded, and sectioned at the University of Ottawa, Department of Pathology and Laboratory Medicine, Histology Core Facility: samples included 10 tumours from PBS-treated mice, 10 tumours from mice intraperitoneally (IP) injected with the ethanol extract of *Uncaria tomentosa*, and 9 tumours IP injected with the PBS extract of *Uncaria tomentosa*. Immunohistochemistry staining for Ki-67 and Factor VIII staining was performed using ABC immunohistochemistry kits as described by the manufacturer (Santa Cruz Biotech, Santa Cruz, CA, USA), [56] and counterstained with haematoxylin. The stained sections were mounted on glass slides in Permount and then analyzed using an Olympus IX3 microscope. Staining was assessed for 8 separate fields for each tumour and a combined score for each mouse was averaged for each treatment group.

For immune markers, the slides were deparaffinized in xylene and hydrated in an ethanol series. The slides were incubated with blocking serum for 1 h (Santa Cruz Biotech) and then with a combination of 3 fluorescently-tagged primary antibodies labelled with Alexa-488, Alexa-546, or Alexa-633 each at a titre of 1:50 (Santa Cruz Biotech.) overnight as described [57]. One slide was incubated with the three negative control antibodies (normal rat IgG2a-Alexa-633, normal mouse IgG2b-Alexa-546, and normal mouse IgG1-Alexa-488) in block buffer to define background fluorescence. A second slide was stained with a combination of antibodies that recognize T cells: anti-CD3-Alexa-546 (all T cells), anti-CD4-Alexa-488 (T helper cells), and anti-CD8-Alexa-633 (cytotoxic T cells). A third slide was incubated with antibodies against the leukocyte markers, anti-CD19-Alexa-549 (B cells) and anti-CD45-Alexa 633 (pan leukocyte). A fourth slide was stained with markers corresponding to endothelial cells and to activated endothelial cells: anti-ICAM-1-Alexa-633, anti-E-selectin-Alexa-546, and anti-PECAM -1-Alexa-488. The slides were washed with PBS, pH 7.4, mounted using UltraCruz^®^ Hard-set Mounting Medium (Santa Cruz Biotech.) and analyzed using an Olympus IX3 fluorescence microscope set for different fluorescent channels (green, red, and far red). The number of cells/hpf (approximately 0.04 mm^2^) was counted, the average of 8 separate fields for each tumour determined, and the mean ± SD reported for each treatment group (*n* = 9 or 10) used for comparison.

### 4.12. TUNEL Staining of Histochemical Sections

For the TUNEL staining of histochemical sections, the slides were deparaffinized, hydrated using an ethanol series, and incubated in blocking buffer (100 mM Tris-HCl, pH 7.5, 20% FBS, and 3% BSA). Each section was stained by incubation with 50 µL of TUNEL reaction mixture (Roche) for 60 min at 37 °C in a humidified atmosphere in the dark. Images were captured using an Olympus IX3 fluorescence microscope for analysis.

### 4.13. Statistical Analysis

Data were expressed as the mean ± SD of at least three independent analyses. The mean values were subjected to a one-way ANOVA followed by a Tukey post hoc analysis to test significant differences (*p* < 0.05) between groups. Comparison between the treatment and control samples for the caspase assays were performed using a Student’s *t*-test.

## Figures and Tables

**Figure 1 molecules-26-01066-f001:**
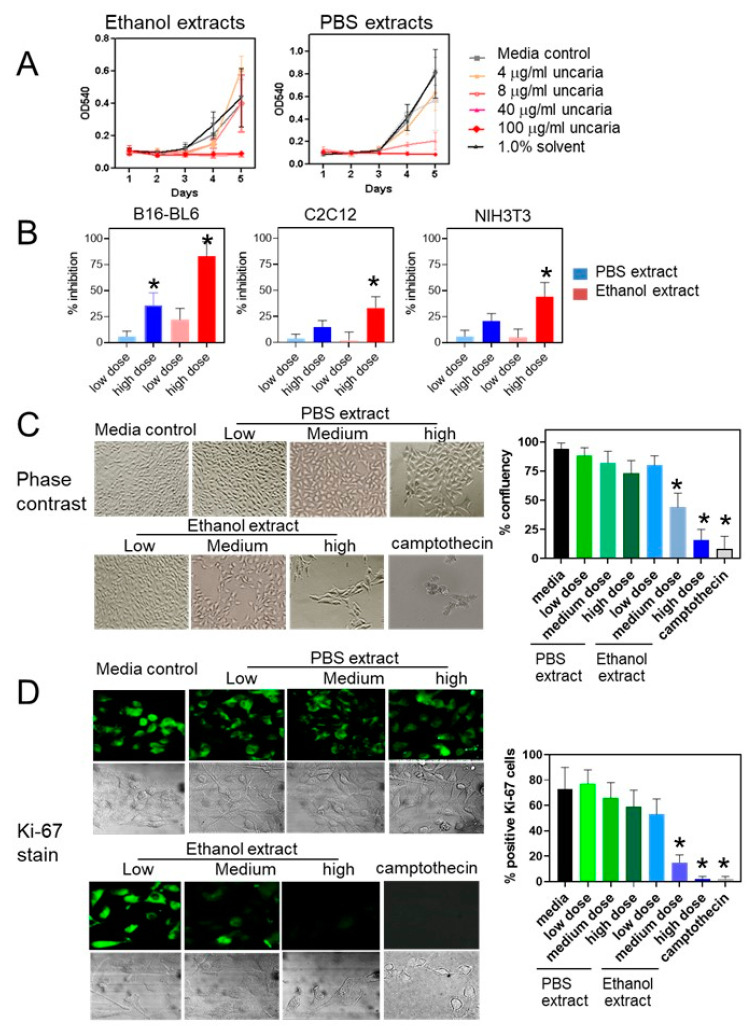
Inhibition of B16- BL6 cell proliferation treated with *Uncaria tomentosa* extracts: (**A**) B16-BL6 melanoma cells were treated with *Uncaria tomentosa* extracted with 70% ethanol or PBS at 4, 40, or 100 μg/mL on day one of treatment and relative cell viability determined every day over a period of 4 days using an MTT assay; (**B**) the relative viability of B16-BL6, NIH3T3, and C2C12 cells was determined for cells treated with *Uncaria tomentosa* extracts for 4 days using an MTT assay and the percent inhibition of cell viability was determined compared to media controls; (**C**) phase contrast micrographs of cell cultures treated with Uncaria extracts for 3 days showing changes in cell confluency as presented in the bar graph; and (**D**) fluorescence microscopic images of Ki-67-stained B16-BL6 cells and phase contrast images were taken on day 3 of treatment with *Uncaria tomentosa* extracts and plotted as the % of cells that were positive for KI-67 staining. Data were analyzed for three independent experiments and differences from control were considered significant (*) when *p* ≤ 0.05 by a Tukey post hoc test.

**Figure 2 molecules-26-01066-f002:**
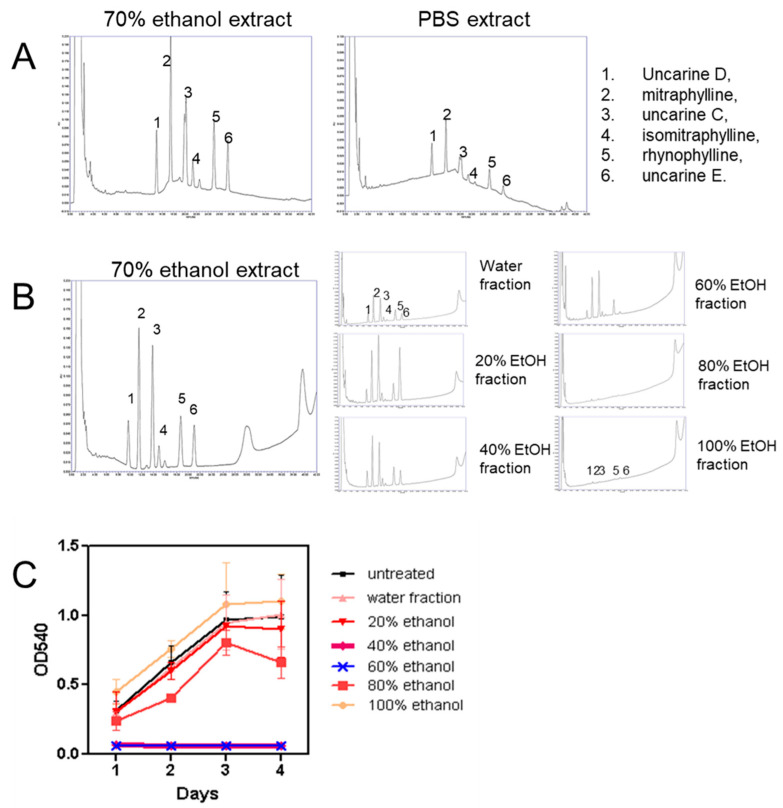
The analysis of *Uncaria tomentosa* extracts using HPLC. *Uncaria tomentosa* was extracted by boiling in (**A**) 70% ethanol or PBS, clarified by filtration and then subjected to HPLC using a C18 column; (**B**) an ethanol extract of Uncaria tomentosa was fractionated on a PVPP column and fractions eluted at 100% water, 20% ethanol, 40% ethanol, 60% ethanol, 80% ethanol, and 95% ethanol were concentrated and subjected to HPLC. The peaks, detected by absorbance at 245 nm, were identified by comparison to the elution of standards for *Uncaria tomentosa* under the same conditions on the same day: (1) uncarine D, (2) mitraphylline, (3) uncarine C, (4) isomitraphyline, (5) rhynophyline, and (6) uncarine E; and (**C**) MTT assays were performed to evaluate the effect of treatment with the PVPP fractions on B16-BL6 viability.

**Figure 3 molecules-26-01066-f003:**
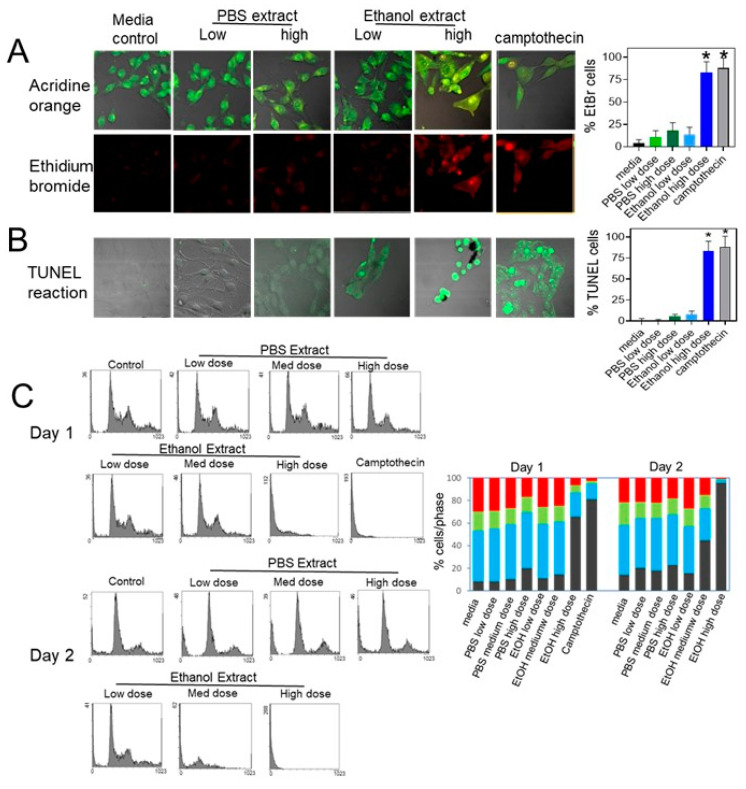
Effect of *Uncaria tomentosa* extracts on the induction of apoptosis by the B16-BL6 cells. B16-BL6 cells were treated with culture media (control), or low (4 μg/mL), medium (40 μg/mL), or high doses (100 μg/mL) of *Uncaria tomentosa* extracted with 70% ethanol or PBS, or with 6 µg/mL Camptothecin (positive control): (**A**) fluorescence images were obtained for cells treated with Uncaria tomentosa extracts for 3 days and then stained with acridine orange (green) or ethidium bromide (red). The bar graph shows the % of cells stained with ethidium bromide; (**B**) cells treated for 2 days were stained by incubation with TUNEL reaction mixture to detect DNA fragmentation and the monolayers were analyzed using a fluorescence microscope and representative photos showing an overlay of the fluorescence channel (excitation at 488 nm and emission at 520 nm) and phase contrast channel are shown. The bar graph shows the % of cells that are positive for DNA fragmentation from three independent experiments. Differences from control were considered significant (*) when *p* ≤ 0.05 by a Tukey post hoc test; (**C**) flow cytometry of B16-BL6 cells treated with extracts of *Uncaria tomentosa* for 24 and 48 h and incubated with propidium iodide was used to determine the proportion of cells with a sub-G1 DNA concentration indicative of induction of apoptosis. The percentage of cells in the sub-G1 (black), G1 (blue), S (green) and G2 phases (red) are shown as the mean of three independent experiments.

**Figure 4 molecules-26-01066-f004:**
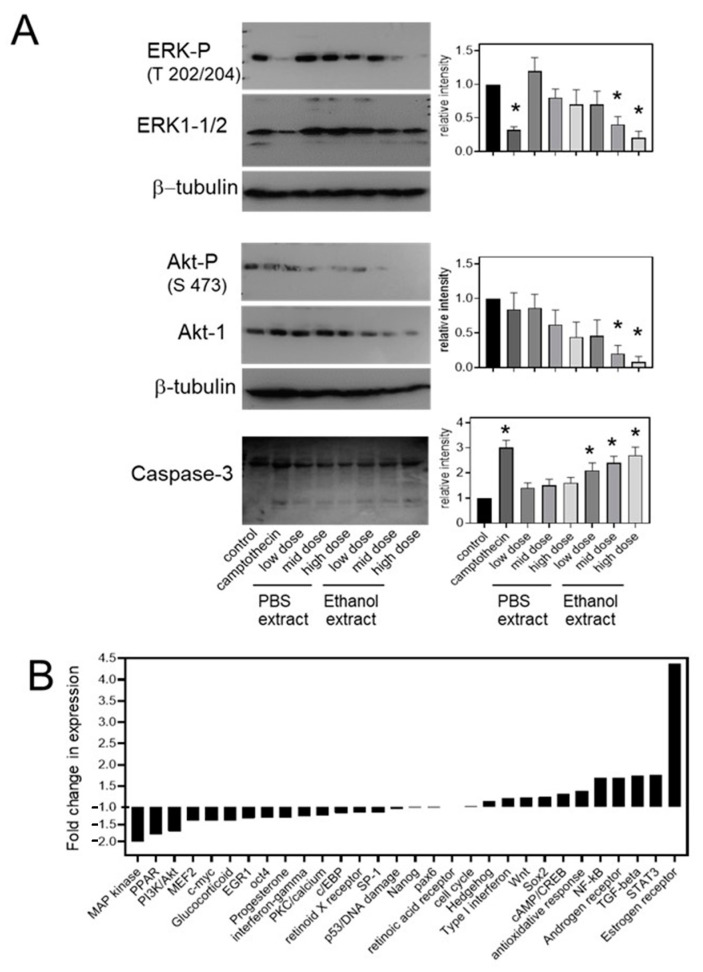
Treatment with *Uncaria tomentosa* inhibits ERK and Akt phosphorylation and promotes caspase cleavage in B16-BL6 cells: (**A**) B16-BL6 cells were treated with ethanol or PBS extracts of Uncaria tomentosa or 10^−6^ M Camptothecin for 48 h and cell lysates were collected and subjected to immunoblot analysis using ERK (threonine 202/204) phosphate, ERK1/2, Akt (serine 473) phosphate, Akt, β-tubulin, and caspase-3 antibodies. The relative intensities of the phosphorylated bands, normalized for the level of total protein or the relative level of cleaved/uncleaved forms, were determined by densitometry and the fold change for 3 independent experiments, as shown in the bar graphs. Differences compared to the control were considered significant (*) when *p* ≤ 0.05 by a Tukey post hoc test; (**B**) the relative activity of the indicated transcription factor pathways for B16-BL6 cells treated with 100 µg/mL of the ethanol extract of *Uncaria tomentosa* for 24 h was determined using the Cignal transcriptional profile assay. The relative luminescence was corrected for a constitutive control plasmid and fold changes between the Uncaria- and vehicle only-treated controls are reported for two experiments.

**Figure 5 molecules-26-01066-f005:**
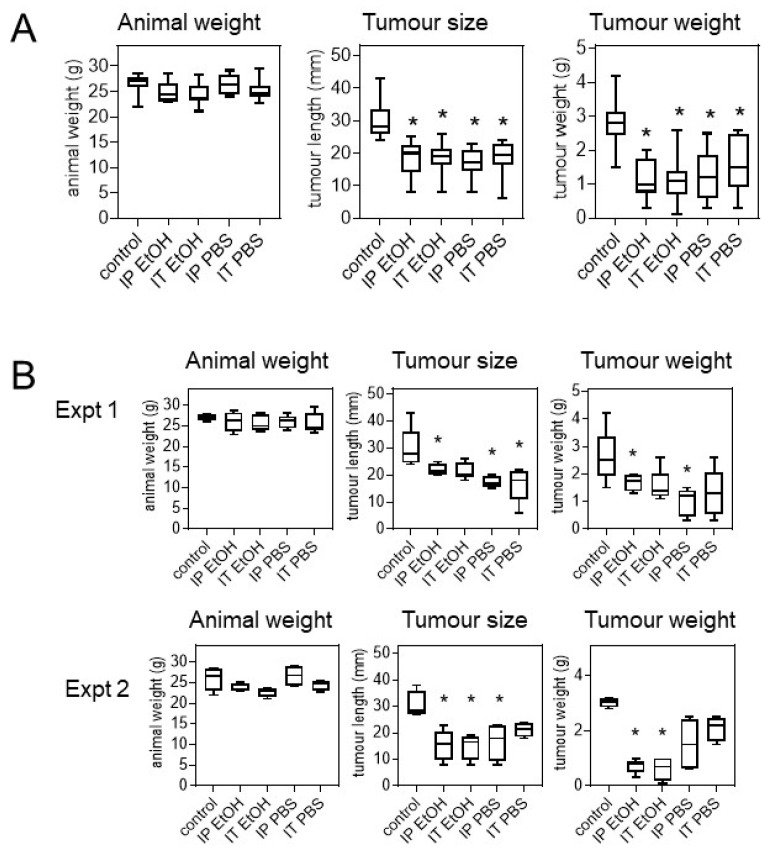
Tumour weight was evaluated following treatment with *Uncaria tomentosa*. Animals were injected with B16-BL6 tumour cells and then received intraperitoneal (IP) or intra-tumour (IT) injections with *Uncaria tomentosa*, extracted with 70% ethanol (EtOH) or PBS, pH 7.4, (PBS) that had been freeze-dried and resuspended in PBS. At endpoint, the mice were euthanised and the tumours were removed to determine the tumour diameters (size) and weights, and the mouse body weights. The average measures for each of the five treatment groups was plotted using a box and whisker plot (whiskers are 95% interval) for the (**A**) combined experiments (*n* = 8–10) or (**B**) two separate experiments (*n* = 4–5). The results revealed that there were significant differences (*p* < 0.05) between each of the four treated groups compared to the control (*) using ANOVA and Tukey post hoc statistical analysis.

**Figure 6 molecules-26-01066-f006:**
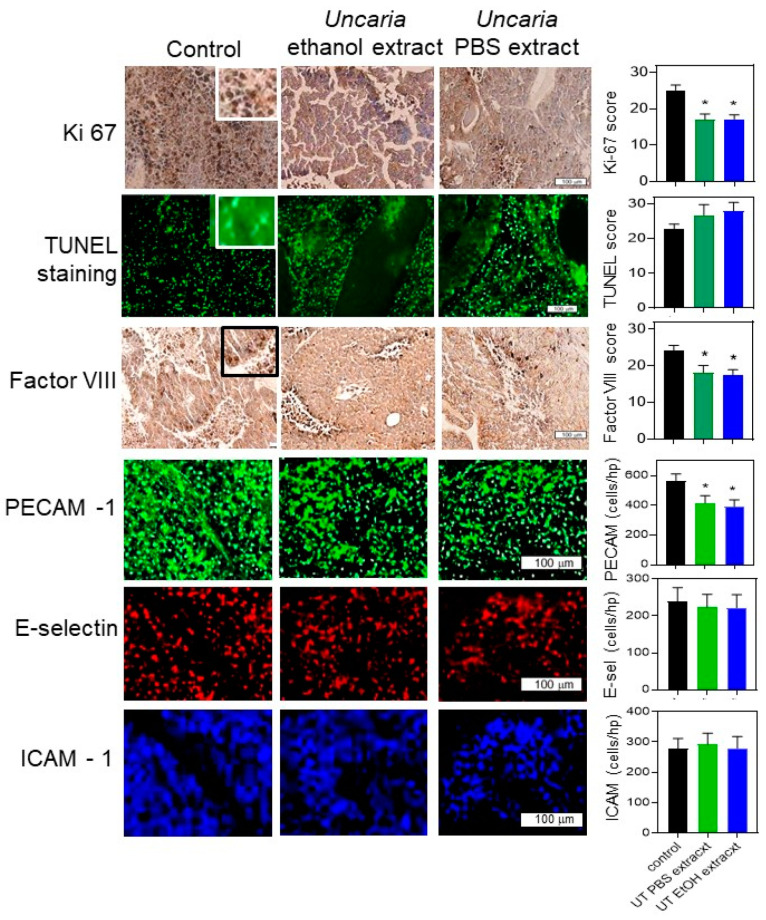
Histochemical analysis of cell viability and endothelial cell markers in sections of B16-BL6 tumours isolated from mice treated with *Uncaria tomentosa* extracts. Histochemical analysis was performed on B16-BL6 tumour tissue from mice treated with the vehicle, the resuspended ethanol extract, or the resuspended PBS extract of *Uncaria tomentosa* and typical microscopic fields shown. Sections were stained with the anti-Ki67 antibody, TUNEL reagent, or anti-Factor VIII antibody and the relative intensity of staining was determined from 8 images/tumour on an ordinal scale (0–5) directly from microscopic images. Inserts in the control images show higher magnification images of stained cells. A tissue section was triple-stained with fluorescence-labelled antibodies against the endothelial cell markers PECAM-1, E-selectin, and ICAM-1. The number of labelled cells was counted for each high-power field (hpf) from 8 different fields and the mean and standard deviation determined for each condition is shown in the bar graphs. One-way ANOVA and post hoc Tukey tests were performed to reveal any significant differences (*) in staining levels between tumours treated with *Uncaria tomentosa* extracts and vehicle only control tumours (*p* < 0.05).

**Figure 7 molecules-26-01066-f007:**
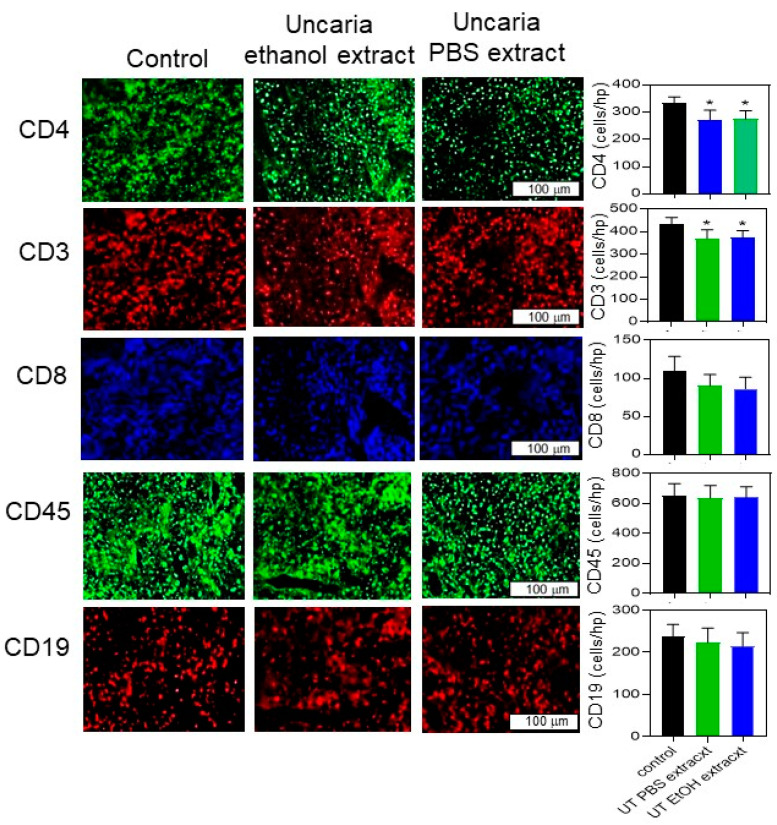
Histochemical analysis of infiltrating immune cells in sections of B16-BL6 tumours isolated from mice treated with *Uncaria tomentosa* extracts. Histochemical analysis was performed on B16-BL6 tumour tissue from mice treated with vehicle, the resuspended ethanol extract, or the resuspended PBS extract of *Uncaria tomentosa* and typical microscopic fields shown. Tissue sections were stained with fluorescence-labelled antibodies against T-cell markers including CD4 (T helper cells), CD3 (pan T cells), and CD8 (cytotoxic T cells), and leukocyte markers including CD45 (pan leukocytes) and CD19 (B cells). The number of labelled cells/hpf was counted from 8 different fields and the mean and standard deviation determined for each condition is shown in the bar graphs. One-way ANOVA and post hoc Tukey analysis were performed to reveal any significant differences (*) in the number of cells between conditions (*p* < 0.05).

## Data Availability

The data presented in this study are available in “Treatment with *Uncaria tomentosa* promotes apoptosis in B16-BL6 mouse melanoma cells and inhibits the growth of B16-BL6 tumours.”

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
