# Peer review of "Treatment with Uncaria tomentosa Promotes Apoptosis in B16-BL6 Mouse Melanoma Cells and Inhibits the Growth of B16-BL6 Tumours"

_molecules, 2021, doi:10.3390/molecules26041066_

Round 1
Reviewer 1 Report
Please see the attachment.

Author Response
Reviewer #1
Several changes have been made to improve the manuscript based on some of the issues raised by the reviewer.
One of the major issues pointed out by the reviewer was that the clarity of some of the figures was not sufficient. Several modifications to both the pictures and the legends to the figures have been made to address these issues. For example, a more clear description of the data shown and to indicate which extracts were used in each graph are now more clear. The statistical tests and the number of repeats was added. Several of the micrographs have been altered to better sow the data. The Ki-67 staining data was redrawn for Figure 1D to separate the fluorescence and phase contrast channels and emphasize the difference in % stained cells. The data in Figure 6 has now been separated into two figures (Figure 6 and Figure 7) to allow the data to be presented in a larger format so that the micrographs can be more easily seen. Scale bars have been added to the images and insets to show some details of the staining have been added to some of the images.
1. The description of the MTT data has been altered to reflect the idea that MTT assays are more correctly referred to as measures of cell viability rather than measures of proliferation. The data in Figure 1 also includes a measure of cell confluency and a more complete presentation of the Ki-67 proliferation antigen data to show the number of cells that are dividing, which supports the MTT data.
The description of the apoptosis data in Figure 3 does include morphological staining with acridine orange and ethidium bromide, TUNEL evaluation of DNA fragmentation, and flow cytometry to detect nuclear fragmentation (sub-G1 particles). The data in Figure 4 also includes a caspase-3 immunoblot. All show a similar pattern of activation of apoptosis supporting the idea that treatment with the ethanol extract is more effective at induing apoptosis in a time- and dose-dependent manner.
2. The rational for choosing the partcular antibodies in the histochemical analysis has been clarified by comparison to previous reports and to measure infiltration of the commonly associated lymphocyte populations in tumour tissues
3, We have more explicitly described the results of the in vivo data to show that biweekly injection of Uncaria extracts decrease the weight of the B16-BL6 tumours by about 60% and that the average diameter of the tumour is decreased by 40%, which are statistically significant.
4. The staining of the tissue samples with TUNEL reagent is more clearly described to indicate scoring for relative number of stained cells rather than just relative staining intensity. The fluorescence background has been subtracted by setting the thresholds levels of the microscopic images based on the control images.
5. The description of the ethanol extract used in the in vivo experiment has been more clearly explained and made more prominent in both the text and figure legends to point out that the extracts had been freeze-dried and resuspended in PBS prior to injection to avoid injection of large amounts of ethanol into the mice.
6. The discussion has been edited to limit the description of results and to focus more on interpretation. Some additional material has been added regarding limitations to interpretation and discussion of the potential for Uncaria in cancer therapy as requested by the other reviewers.
Minor points
The Figure 1C description has been clarified to emphasize that the images were taken using phase contrast microscopy to measure monolayer confluence rather than subtle changes in morphology.
Figure 1D has been redrawn to separate the fluorescence and phase contrast images of the Ki-67 data and the graph altered to show the data as percent positive Ki-67 stained cells (rather than relative staining) which helps to show the changes in cell proliferation in response to Uncaria treatment.
The cell cycle data in figure 3C is now presented as a cumulative bar graph to include a measure of all phases of the cell cycle.
The phosphorylated residues for ERK (T 202/20) and Akt (S 473) are now included in the text and figures and a panel representing the b-tubulin loading control for each blot has been added (although the densitometry is still calculated as a relative measure of ERK-P divided by ERK1/2 band intensity) .
The highest dose of the ethanol extract of Uncaria (100ug/ml) was used for 24 h for the transcriptional profiling array since this concentration was able to induce strong effects at 24 h which is prior to complete destruction of the cells by most of our assays.
As above, the images in Figure 6 have been enlarged by splitting the figure into two allowing a better presentation of the data. More clear labelling of the cells is included. However, we did not counterstain these particular slides with hematoxylin since that diminished the fluorescence signal. Therefore, staining of separate sections would have to have been presented as a measure of the cellularity of the sections and this is already presented in the ABC-stained samples for the Ki-67 and Factor VIII samples which were obtained from serial sections of the same tumour blocks.
Reviewer 2 Report
The study entitled “Treatment with Uncaria tomentosa promotes apoptosis in B16-BL6 mouse melanoma cells and inhibits the growth of B16-BL6 tumours” describes the anti-melanoma effects of aqueous (PBS) and ethanolic extracts of U. tomentosa (cat’s claw) in vitro (BL16-BL6 cell line) and in vivo (B16-BL6 melanoma/C57BL/6 isogenic tumour transplantation model). Results suggest antiproliferation and apoptotic effects (including caspase 3 cleavage), and inhibition of ERK and PI3K/Akt pathways.
The work presented is not entirely novel and, as mentioned in the introduction of this study, apoptotic effects have been reported in multiple cancer cell lines. In addition, this study is partially covered by other studies, e.g.:
- Caballero et al. 2005, Acta Científica Venezolana 56(1):32-36 reported that the aqueous extract of U. tomentosa did not present significant cytotoxic effects in B16/BL6 cells (3 mg/ml, 24h, MTS assay). Nevertheless, the number of metastasis to the lung was found decreased in the B16-BL6 melanoma/C57BL/6 isogenic tumour transplantation model.
- Fazio et al. 2008 (Boletín Latinoamericano y del Caribe de Plantas Medicinales y Aromáticas, 7 (5), 217 – 224) showed that the ethanolic extract of U. tomentosa inhibited B16/BL6 cell viability (0.3 mg/ml, 24h, MTS assay) and in the B16-BL6 melanoma/C57BL/6 isogenic tumour transplantation model, a reduction in tumor size was observed.
- Lozada-Requena et al. 2015, Rev Peru Med Exp Salud Publica;32(4):633-42 showed that in the B16-BL6 melanoma/C57bl/6 isogenic tumour transplantation model, the administration of ethanolic extract of U. tomentosa did not result in significant alterations in CD4/CD8 ratio, as well as no significant alteration in levels of CD3+ lymphocyte population.
In the present study, a few points require the authors' careful attention, as they raise concerns about the experimental design, results and conclusions:
- Fig. 1. The legend is missing for panel 1D
- Section 4.8: immunoblot analysis – no loading control is presented in materials and methods (lines 615-632) and in results (Fig. 4) – this control is essential for results validation
- Section 4.9: Signalling pathways – in materials and methods a very high extract concentration is mentioned (100 mg/ml) – the authors must clarify why this high concentration was used.
- Histochemical analysis (Fig. 6) – concerning the markers Factor VIII, TUNEL, PECAM-1, ICAM-1 there are discrepancies between the values presented on the side bar charts and the ones reported in Dr Zari’s doctoral thesis of (https://core.ac.uk/download/pdf/222897738.pdf): Factor VIII page 159 (Fig. 3.34B) vs this study Factor VIII (Fig. 6); TUNEL page 161 (Fig. 3.35) vs this study TUNEL (Fig. 6); PECAM-1 and ICAM-1 page 176 (Fig. 3.39) vs. this study. The authors must present a reason for these small differences in the bar charts, particularly since in this study significant differences are pointed that are not reported in the original doctoral thesis
Author Response
Reviewer #2
The reviewer has pointed out some additional studies that looked at the effect of Unacaria tomentosa on B16-BL6 tumour growth. We have added the new references and included a better introduction section on how these studies were relevant to the current report.
Figure 1D has been rewritten to clarify the description of the data (especially as Figure 1D has also been altered).
Loading controls (b-tubulin) have been added to Figure 4A (although the densitometry is still based on the ratio of ERK-P/ERK1/2 band intensities).
The description of 100 mg/ml Uncaria in Figure 4B is a mistake and has been corrected to 100 ug/ml (the high dose of Uncaria used in the other experiments).
The reviewer has pointed out that the results of the immunohistochemistry analysis are somewhat different from the way they are described in Dr Zari’s thesis. In his thesis, Dr Zari based his analysis on assigning intensity according to an ordinal Likert scale (0-5). While this showed some small difference in relative staining, these differences were not quite significant. In the current report, the number of fluorescent cells in the micrograph were counted (by R.L.) and reported as cells per high power field. This change in the measurement of cell infiltration showed similar tends in the data, but the difference between the “granularity” of the 0-5 Likert scale and cell counts revealed that these small differences were statistically significant for some of the cell counts. The bar graphs look very similar between the two methods of analysis although the specific values (and therefore significance) are different.
Reviewer 3 Report
This is an interesting experimental study of UT on melanoma cells. Some points could improve the paper.
Minor points:
- Abstract and elsewhere. Replace abbreviations by expanded words when mentioned first. See abstr PBS, as example.
- Add in the abstract actual numbers with SEM or SD and p values for a few parameters that are most important.
- Introduction is lenghty, condense. Include some clinical data on human melanoma. treatment, and outcome. Inculde in the text, why novel therapy modalities are needed.
- Provide more details that UT is well tolerated in humans and expand statements on side effects. Are there cases of HILI like in many cases of treatment by herbs?
- Includde in the discussion what studies should be done in patients with melanoma. RCTs? How should UT be applied? Is it available as drug? IV? local application? Against what standard therapy should UT be tested?
Author Response
Reviewer #3
Changes to improve the manuscript based on the comments of the reviewer have been made. The text has been altered to properly format the presentation of abbreviations.
The abstract has been modified to include some of the actual values and standard deviations for some of the important variable as suggested. For example, the decrease in tumour weight (59+13) and diameter (40+11) have been included.
We have tried to make the introduction more concise and focused by reordering and editing some of the sections. However, we have also added some material related to Uncaria treatment in other animal and clinical trial models to the introduction as suggested by the reviewers.
A few comments about how Uncaria tomentosa might be relevant to clinical use in the therapy of patients with melanoma have been added to the end of the discussion to indicate the potential relevance of this study.
Round 2
Reviewer 1 Report
Although authors did made some suggested corrections and changes (consisting mainly of previous data recalculation), none of the suggested experiments were perform. Therefore the huge part of immunofluorescence results are still not convincing.
Author Response
The reviewer is correct that no new experiments were performed as part of the revision of the manuscript. We have attempted to address the comments by redrawing the figures to make the immunofluorescence more clear and to better describe how the cell counts were obtained. It is our hope that this meets the best standard for presentation of the data in that the pictographs support the quantitative assessment of the cell counts in the various conditions.
Reviewer 2 Report
The revised version of the manuscript titled "Treatment with Uncaria tomentosa promotes apoptosis in B16-BL6 mouse melanoma cells and inhibits the growth of B16-BL6 tumours" contains minor aspects that nevertheless require the authors' attention, e.g.:
- Line 200 of the revised version: 6 mg/ml Camptothecin is too high concentration compared to the one typically used in apoptosis controls. This concentration doesn’t seem to match the 1 micromolar concentration ~ 0.35 microgram/ml Camptothecin (line 245 of this version) cited for protein work. Was the actual concentration (line 200) 6 microgram/ml Camptothecin?
- Line 745: One author is misspellelt: Arseak should be Arsenak
- Lines 789, 790: incorrect author name and volume/page numbers. The correct reference is: Caballero, M., Arsenak, M., Abad, M.J., Cesari, I.M., Taylor P.G. Effect of 3 plant extracts on B16-BL6 melanoma cell growth and metastasis in C57BL/6 mice. Acta Cientifica Venezolana 2005, 56, 32—36
Author Response
We thank the reviewer for finding these errors and have corrected them.
Camptothecin was used at 6 ug/ml in the experiments (not 6 mg/ml) and this change has been made.
The references, #8 and #25 have been corrected to properly spell the author's names and to correct the volume and page numbers.